# BF₃–Catalyzed Diels–Alder Reaction between Butadiene and Methyl Acrylate in Aqueous Solution—An URVA and Local Vibrational Mode Study

**Marek Freindorf †and Elfi Kraka \*,†**

Department of Chemistry, Southern Methodist University, Dallas, TX 75275, USA; mfreindorf@smu.edu
* Correspondence: ekraka@smu.edu
† These authors contributed equally to this work.

**Abstract:** In this study we investigate the Diels–Alder reaction between methyl acrylate and butadiene, which is catalyzed by BF₃ Lewis acid in explicit water solution, using URVA and Local Mode Analysis as major tools complemented with NBO, electron density and ring puckering analyses. We considered four different starting orientations of methyl acrylate and butadiene, which led to 16 DA reactions in total. In order to isolate the catalytic effects of the BF₃ catalyst and those of the water environment and exploring how these effects are synchronized, we systematically compared the non-catalyzed reaction in gas phase and aqueous solution with the catalyzed reaction in gas phase and aqueous solution. Gas phase studies were performed at the B3LYP/6-311+G(2d,p) level of theory and studies in aqueous solution were performed utilizing a QM/MM approach at the B3LYP/6-311+G(2d,p)/AMBER level of theory. The URVA results revealed reaction path curvature profiles with an overall similar pattern for all 16 reactions showing the same sequence of CC single bond formation for all of them. In contrast to the parent DA reaction with symmetric substrates causing a synchronous bond formation process, here, first the new CC single bond on the CH₂ side of methyl acrylate is formed followed by the CC bond at the ester side. As for the parent DA reaction, both bond formation events occur after the TS, i.e., they do not contribute to the energy barrier. What determines the barrier is the preparation process for CC bond formation, including the approach diene and dienophile, CC bond length changes and, in particular, rehybridization of the carbon atoms involved in the formation of the cyclohexene ring. This process is modified by both the BF₃ catalyst and the water environment, where both work in a hand-in-hand fashion leading to the lowest energy barrier of 9.06 kcal/mol found for the catalyzed reaction **R1** in aqueous solution compared to the highest energy barrier of 20.68 kcal/mol found for the non-catalyzed reaction **R1** in the gas phase. The major effect of the BF₃ catalyst is the increased mutual polarization and the increased charge transfer between methyl acrylate and butadiene, facilitating the approach of diene and dienophile and the pyramidalization of the CC atoms involved in the ring formation, which leads to a lowering of the activation energy. The catalytic effect of water solution is threefold. The polar environment leads also to increased polarization and charge transfer between the reacting species, similar as in the case of the BF₃ catalyst, although to a smaller extend. More important is the formation of hydrogen bonds with the reaction complex, which are stronger for the TS than for the reactant, thus stabilizing the TS which leads to a further reduction of the activation energy. As shown by the ring puckering analysis, the third effect of water is space confinement of the reacting partners, conserving the boat form of the six-member ring from the entrance to the exit reaction channel. In summary, URVA combined with LMA has led to a clearer picture on how both BF₃ catalyst and aqueous environment in a synchronized effort lower the reaction barrier. These new insights will serve to further fine-tune the DA reaction of methyl acrylate and butadiene and DA reactions in general.

**Keywords:** Diels–Alder; URVA; local mode analysis; QM/MM; catalysis

## 1. Introduction

The Diels–Alder (DA) reaction is a prototypical example of a concerted, stereospecific, pericyclic symmetry-allowed [4 + 2] cycloaddition, which has been widely studied by experiment and theory [1]. The DA reaction is frequently used in organic synthesis to obtain six-membered functionalized carbocyclic compounds and the total synthesis of natural products [2–4], in industry [5,6], medicine [7,8], materials science [9–14] and biochemistry [15–21]. Because of the tremendous impact of the DA reaction on opening new perspectives in organic chemistry and becoming a fundamental tool in chemical synthesis, Diels and Alder were awarded for their discovery with the Nobel Prize in Chemistry in 1950.

The DA reaction involves the symmetry-allowed cycloaddition of a conjugated $4\pi$– electron rich diene with a $2\pi$–electron deficient dienophile following the endo rule [22]. The original study by Woodward suggested a synchronous, i.e., simultaneous, bond formation process generating the resulting six-membered cyclohexene [23]. However, subsequent theoretical studies showed that a synchronous bond formation only takes place in the case of symmetric substrates [24], while the presence of activating groups/substituents leads to asymmetric substrates resulting in a stepwise bond formation [25–31].

Using frontier molecular orbitals (FMO) Woodward [23] and later Houk and others [32–35] showed that the activation energy of the DA reaction can be decreased by decreasing the energy gap between the HOMO of the diene and the LUMO of the dienophile. This can be achieved, e.g., by the use of Lewis acids and/or electron–withdrawing substituents on the dienophile, making it more electrophilic. The decrease of the activation energy by applying this strategy was confirmed by a number of experimental [36–38] and theoretical studies [39–44]. Breslow suggested that DA reactions can be accelerated in aqueous solution by a hydrophobic effect [45]. This effect has been intensively investigated experimentally [46–57] and theoretically [58–70]. Recently, the acceleration of DA reactions in water micro-droplets via a so-called nano-confinement effect was reported. It was further speculated that the catalytic effect of water leading to lower reaction barriers could also be related to a global polarization of the reaction complex along the reaction path and to a stabilization of a transition state by hydrogen bonds [71,72].

In order to shed more light into the catalytic effect of a Lewis acid being attached to the diene and the catalytic effect of aqueous solution, in our study we investigated the DA reaction of methyl acrylate and butadiene in gas phase and aqueous solution, with and without $BF_3$ as Lewis acid catalyst attached to the CO oxygen atom of methyl acrylate. The calculations were performed starting from four different orientations of the reacting species (**R1**, **R2**, **R3**, and **R4** as shown in Figure 1) leading in total to 16 reactions. The reactions are labeled as **R1cw**, **R2cw**, **R3cw**, **R4cw**, **R1nw**, **R2nw**, **R3nw**, **R4nw**, **R1cg**, **R2cg**, **R3cg**, **R4cg**, **R1ng**, **R2ng**, **R3ng**, and **R4ng** where the symbols **c** and **n** indicate catalyzed and non-catalyzed reaction, respectively and the symbols **w** and **g** indicate reaction in aqueous solution and in the gas phase, respectively. We used the Unified Reaction Valley Approach (URVA) [73] following the reaction complex (i.e., the union of diene and dienophile) along the reaction path from entrance to exit channel, complemented by Local Vibrational Mode Analysis (LMA) [74], NBO analysis [75], Bader's quantum theory of atoms-in-molecules (QTAIM) analysis [76,77] and the ring puckering analysis of Cremer and Pople as analytical tools [78,79].

**Figure 1.** Schematic representation of the relative orientation of methyl acrylate and butadiene in the reactions **R1**, **R2**, **R3**, and **R4** catalyzed by BF$_3$, which were investigated in our study.

## 2. Materials and Methods

A comprehensive review of the URVA methodology can be found in Ref. [73], the theoretical basis and background of URVA has also been thoroughly described in Refs. [80–82] Selected URVA applications are discussed, e.g., in Refs. [81,83–92] URVA requires a representative reaction path (usually the intrinsic reaction coordinate (IRC) of Fukui is chosen [93]) which is followed by the reaction complex on its way from the entrance channel, via the TS to the exit channel on the potential energy surface (PES) of the reaction. As the reaction proceeds, the electronic structure of the reaction complex changes, which is registered by vibrational modes perpendicular to the long amplitude motion along the reaction path spanning the so-called Reaction Valley [94–96]. As described in the seminal Miller, Handy, and Adams paper on the Reaction Path Hamiltonian, these vibrational modes couple with the translational motion along the reaction path and the sum of all coupling elements define the scalar reaction path curvature [97]. Hence, any chemical change is reflected by changes in the scalar reaction path curvature. We further decompose the scalar reaction path curvature into components, such as internal coordinates representing bond lengths, bond angles, and dihedral angles, as well as puckering coordinates or pyramidalization angles [80], which allows us to assign the reaction path curvature maxima to specific events occurring along the reaction path, such as bond formation/cleavage processes, atomic rehybridization conformational changes and so on, providing a detailed insight into the reaction mechanism. If the value of a component is positive, it supports the chemical change taking place, if it is negative it resists. In contrast to the reaction path curvature peaks, the curvature minima correspond to minimal electronic structure changes of the reaction path, often reflecting the beginning of a new chemical event. In this way, curvature minima and maxima can be used to divide the entire reaction path into chemically meaningful reaction phases [91,92], defining the characteristic fingerprints of the reaction. Only chemical events occurring before the TS contribute to the energy barrier, so that a careful analysis of the curvature peaks in the entrance channel provides important insights on how to lower a barrier [73].

The theoretical background of LMA, originally developed by Konkoli and Cremer [98–102] can be found in a comprehensive review article [74]. The local vibrational modes of a molecule can be considered as the local counterparts of normal vibrational modes, which are generally delocalized due to electronic- and mass-coupling [103]. Therefore, associated normal mode stretching force constants are of limited use as individual bond strength descriptors. In con-

trast, local vibrational stretching force constants derived from uncoupled local vibrational modes directly reflect the intrinsic strength of a chemical bond and/or weak chemical interaction [104]. We have successfully applied local stretching force constants to characterize covalent bonds [104–111], and weak chemical interactions such as halogen bonds [112–117], chalcogen bonds [118–120], pnicogen bonds [121–123], tetrel bonds [124], and hydrogen bonds [125–133], as well as so-called $\pi$–hole interactions [134]. A one-to-one relationship exists between a complete set of non-redundant local modes and the normal modes via an adiabatic connection scheme [135], forming the characterization of normal mode (CNM) procedure [101]. CNM decomposes each normal mode into local mode contributions and was recently successfully applied to characterize the usefulness of vibrational Stark effect probes [136].

Instead of describing the bond strength via the local mode force constant $k^a$ obtained from LMA, it is chemically more intuitive to describe the bond strength via a bond strength order ($BSO$). Both are connected via a power relationship according to the generalized Badger rule with $BSO = A * (k^a)^B$ [91,106]. The parameters $A$ and $B$ are obtained from two reference molecules with known $BSO$ and $k^a$ values and the requirement that for a zero force constant $k^a$ the corresponding $BSO$ is also zero. For the analysis of the CC bonds we used the CC single bond of ethane ($R(CC) = 1.529$ Å, $k^a = 4.008$ mDyn/Å, $BSO = 1$) and the CC double bond of ethylene ($R(CC) = 1.325$ Å, $k^a = 9.551$ mDyn/Å, $BSO = 2$) as references in our study, leading to $A = 0.3302$ and $B = 0.7982$ at the B3LYP/6-311+G(2d,p) level of theory. For the hydrogen-bond analysis we used the HF bond ($R(HF) = 0.924$ Å, $k^a = 9.437$ mDyn/Å, with $BSO = 1$) and the HF bond in $F_2H^-$ ($R(HF) = 1.148$ Å, $k^a = 0.941$ mDyn/Å, with $BSO = 0.5$) leading to values $A = 0.5092$ and $B = 0.3007$ at the B3LYP/6-311+G(2d,p) level of theory.

Bader's quantum theory of atoms-in-molecules (QTAIM) [76,77,137,138] forms the theoretical basis for identifying, analyzing, and characterizing chemical bonds and interactions via the topological features of the total electron density $\rho(\mathbf{r})$. In this work we used QTAIM as a complementary tool to LMA to assess the covalent character of the CC bonds via the Cremer–Kraka criterion [139–141] of covalent bonding. The Cremer–Kraka criterion is composed of two conditions: necessary condition—(i) existence of a bond path and bond critical point $\rho$, i.e., (3, −1) saddle point of electron density $\rho(\mathbf{r})$ between the two atoms under consideration; (ii) sufficient condition—the energy density $H\rho$ at that point is smaller than zero. $H(\mathbf{r})$ is defined as:

$$H(\mathbf{r}) = G(\mathbf{r}) + V(\mathbf{r}) \tag{1}$$

where $G(\mathbf{r})$ is the kinetic energy density and $V(\mathbf{r})$ is the potential energy density. A negative $V(\mathbf{r})$ corresponds to a stabilizing accumulation of density whereas the positive $G(\mathbf{r})$ corresponds to depletion of electron density [140]. As a result, the sign of $H_c$ indicates which term is dominant [141]. If $H\rho < 0$, the interaction is considered covalent in nature, whereas $H\rho > 0$ is indicative of electrostatic interactions.

The DA reaction involves the formation of a six-membered ring, which preferentially adapts a chair, boat, and tboat conformation, or a mixture of these forms. To analyze the ring conformation of the stationary points of the reactions investigated in this work in more detail, we applied puckering analysis, which is based on the Cremer and Pople puckering coordinates [78,79,142] and has been frequently used in molecular deformation analyses [79,143–156]. For a six-membered ring, the puckering coordinates are split up into the pseudorotational coordinate pair $q_2$ and $\phi_2$ describing the pseudorotation of boat and tboat forms, and a single puckering amplitude $q_3$, which describes the chair form and the inverted chair form. The formula providing the percentage of the chair, boat, and tboat forms can be found in the Supplementary Materials.

The stationary points and IRC calculations in the gas phase were performed at the B3LYP/6-311+G(2d,p) level of theory [157–163]. The energetics of the gas phase reactions were additionally recalculated by performing single point energy calculations at the DLPNO-CCSD(T)/def2-TZVP level of theory [164,165], based on DFT geometries and applying thermochemical corrections from the DFT calculations. The calculations in aqueous solution were performed using the QM/MM methodology, according to the following procedure. The reaction complex, which included butadiene and methyl acrylate with or

without the $BF_3$ catalyst, was surrounded by a TIP3P [166] water sphere of a radius 20 Å with an external harmonic potential of a force constant 10 kcal/mol/$Å^2$. Initial equilibration of the molecular system was done by molecular dynamics (2000 steps of minimization, 100 ps heating from 0 K to 300 K, followed by a production dynamics for 1 ns), followed by an annealing dynamics which was performed for 100 ps by short-time heating the system to 400 K and long-time cooling to 0 K. All simulations were performed on a molecular mechanical level of theory using AMBER [167], with a constrained transition state geometry of the solute. After the annealing dynamics, the molecular system was divided into a QM part which included the reaction complex, and an MM part which included all water molecules. The IRC QM/MM calculations were performed without geometrical constraints using the B3LYP/6-311+G(2d,p)/AMBER level of theory [157–163] with ONIOM [168] starting from the final coordinates obtained from the annealing dynamics. The IRC calculations were performed with Gaussian [169], the gas phase single-point energy calculations with DLPNO-CCSD(T) were done using ORCA [170], and the LMA analysis was done with the LModeA program [171]. The atomic charges along the reaction path were calculated using the NBO program [75], the electron and the QTAIM analysis was performed with the AIMALL program [172]. The puckering analysis was done with the Ring program [173].

## 3. Results

### 3.1. Energetics

The activation energy $E_a$, the reaction energy $E_R$, the activation enthalpy $H_a$, and the reaction enthalpy $H_R$ of the reactions investigated in our study are shown in Table 1, and the corresponding energy profiles are presented in Figure 2. The following discussion refers to the energy values obtained from the DFT calculations. According to Table 1, the smallest activation energy (9.06 kcal/mol) is observed for the catalyzed reaction in aqueous solution **R1cw**, for the initial orientation **R1** of the reacting species (see Figure 1). Therefore in the following, we focus on the details of reaction **R1cw** in comparison with the same reaction in aqueous solution without catalyst **R1nw**, and with the catalyzed **R1cg** and non-catalyzed **R1ng** reaction in the gas phase. The results for corresponding **R2–R4** reactions are summarized in the Supplementary Materials, which also provides for further illustration of the Ball and Stick representations of the reactants, TSs, and products for all 16 reactions investigated in this work as well as reaction videos following the geometry changes of the reaction complex along the reaction path from the entrance to the exit channel.

The activation energy of reaction **R1** in aqueous solution without catalyst **R1nw** is 15.71 kcal/mol, which indicates that the $BF_3$ catalyst lowers the activation energy of the reaction **R1** in aqueous solution by 6.65 kcal/mol. The activation energy of the same reaction **R1** catalyzed in the gas phase **R1cg** is 15.61 kcal/mol, showing that the water environment lowers the activation energy of the catalyzed reaction **R1** by 6.55 kcal/mol. The activation energy of the non-catalyzed reaction **R1** in the gas phase, i.e., reaction **R1ng**, has the largest value (20.68 kcal/mol) in this series. In conclusion, the combined effect of both the $BF_3$ catalyst and the water environment, lowers the activation energy of the reaction **R1** by 11.62 kcal/mol (difference between $E_a$ values of **R1ng** and **R1cw**). While the energetics provide important overall effects, they do not disclose the actual reaction mechanism, in particular answering the important question how the effects of catalyst and aqueous solution are synchronized. This will be explored in the following based on the URVA analysis.

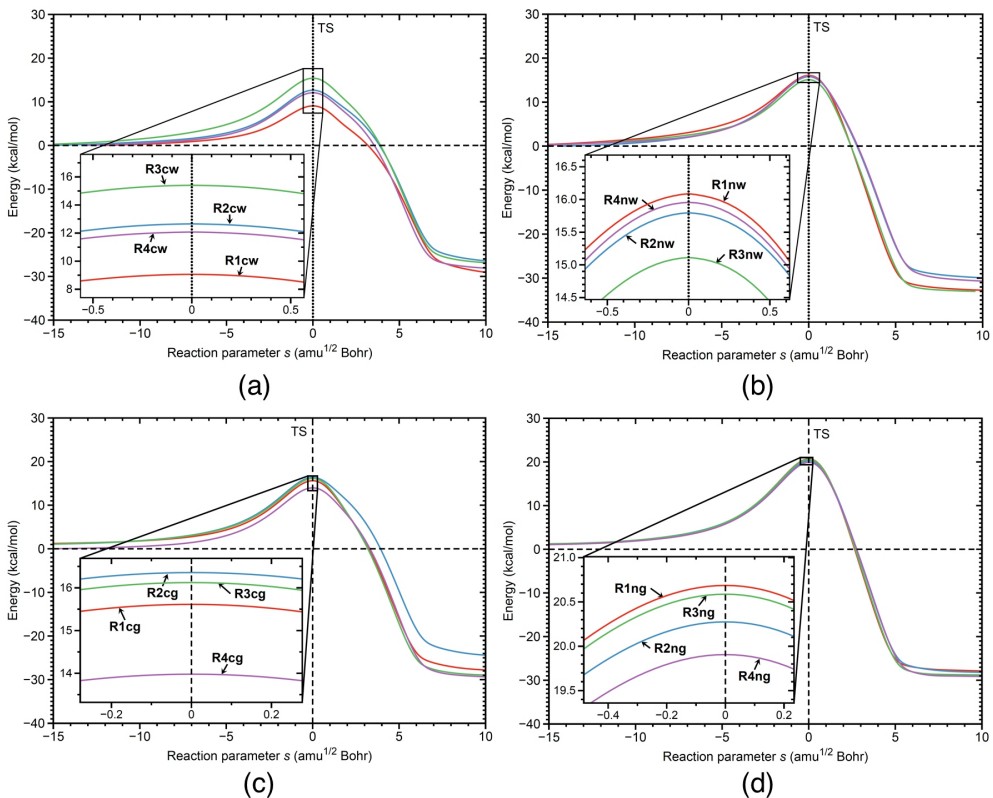

**Figure 2.** Energy profiles for: (**a**) catalyzed reactions in water solution **R1cw**, **R2cw**, **R3cw**, and **R4cw**; (**b**) non-catalyzed reactions in water solution **R1nw**, **R2nw**, **R3nw**, and **R4nw**; (**c**) catalyzed reactions in the gas phase **R1cg**, **R2cg**, **R3cg**, and **R4cg**; (**d**) non-catalyzed reactions in the gas phase **R1ng**, **R2ng**, **R3ng**, and **R4ng**. B3LYP/6-311+G(2d,p)/AMBER level of theory for aqueous solution and B3LYP/6-311+G(2d,p) level of theory for gas phase calculations.

**Table 1.** Activation energy $E_a$, reaction energy $E_R$, activation enthalpy $H_a$, and reaction enthalpy $H_R$ of the reactions investigated in our study [a]. The B3LYP/6-311+G(2d,p)/AMBER level of theory for water solution. Gas phase: B3LYP/6-311+G(2d,p) level of theory, and DLPNO-CCSD(T)/def2-TZVP level of theory with thermochemical corrections from the DFT calculations.

| Reaction | Energy | | Enthalpy | | Reaction | Energy | | Enthalpy | |
|---|---|---|---|---|---|---|---|---|---|
| | $E_a$ | $E_R$ | $H_a$ | $H_R$ | | $E_a$ | $E_R$ | $H_a$ | $H_R$ |
| DFT | | | | | | | | | |
| **R1cw** | 9.06 | −30.56 | 9.03 | −27.39 | **R1nw** | 15.71 | −33.24 | 15.83 | −30.31 |
| **R2cw** | 12.65 | −29.94 | 12.40 | −28.03 | **R2nw** | 15.42 | −30.46 | 15.22 | −27.75 |
| **R3cw** | 15.40 | −27.46 | 14.92 | −24.82 | **R3nw** | 14.74 | −33.39 | 14.45 | −30.93 |
| **R4cw** | 12.08 | −29.60 | 11.92 | −27.49 | **R4nw** | 15.96 | −31.28 | 15.69 | −28.02 |
| **R1cg** | 15.61 | −34.26 | 15.41 | −31.38 | **R1ng** | 20.68 | −34.61 | 20.43 | −31.73 |
| **R2cg** | 16.35 | −32.77 | 16.17 | −29.94 | **R2ng** | 20.27 | −34.24 | 20.02 | −31.43 |
| **R3cg** | 16.12 | −29.19 | 15.96 | −26.25 | **R3ng** | 20.59 | −28.82 | 20.33 | −25.96 |
| **R4cg** | 13.97 | −29.69 | 13.95 | −26.71 | **R4ng** | 19.91 | −29.16 | 19.67 | −26.31 |
| CCSD(T) | | | | | | | | | |
| **R1cg** | 15.96 | −43.32 | 15.76 | −40.44 | **R1ng** | 18.28 | −44.89 | 18.03 | −42.01 |
| **R2cg** | 16.40 | −41.56 | 16.22 | −38.73 | **R2ng** | 17.83 | −44.69 | 17.57 | −41.88 |
| **R3cg** | 16.74 | −38.60 | 16.58 | −35.66 | **R3ng** | 19.20 | −38.49 | 18.94 | −35.63 |
| **R4cg** | 15.82 | −38.98 | 15.80 | −35.99 | **R4ng** | 18.89 | −38.43 | 18.65 | −35.58 |

[a] The values are relative to the van der Waals complex of the reactants of the corresponding reaction, and they are given in kcal/mol. The experimental activation energy of the non-catalyzed reaction **R1** in benzene is 18.0 ± 1.0 kcal/mol [36].

### 3.2. Reaction Mechanism

In the following, the main features of the URVA analysis for the parent DA reaction between 1,3 butadiene and ethene are presented (Figure 3), followed by the discussion of reactions **R1cw** (Figure 4), **R1nw** (Figure 5), **R1cg** (Figure 6) and **R1ng** (Figure 7). The corresponding URVA plots for reactions **R2cw**, **R3cw**, **R4cw**, **R2nw**, **R3nw**, **R4nw**, **R2cg**, **R3cg**, **R4cg**, **R2ng**, **R3cg**, and **R4cg** are collected in the Supplementary Materials.

Figure 3 summarizes the main features of the URVA analysis for the parent DA reaction between 1,3 butadiene and ethene, which we extensively discussed in earlier work [24,80,90,92]. As disclosed by the curvature profile shown in Figure 3a, in the parent DA reaction both new CC bonds are simultaneously formed due to the symmetry of the parent DA reaction complex, i.e., the $C_aC_f$ and the $C_bC_c$ components (red line in Figure 3) overlap. The bond-forming event occurs far out in the exit channel, denoted by the curvature peak at $s = 5$ path units. There are only small curvature enhancements in the entrance channel and around the TS, providing another example that although the TS is an energetically important point, it may not be mechanistically relevant [91,92]. Shortly after the TS electron spin, decoupling of the $\pi$-bonds leads to aromatic CC bond equalization (at $s = 0.5$ path units the bond length of all butadiene and ethylene CC bonds ($C_dC_e$ purple solid line; $C_aC_b$, solid green line; $C_eC_f$, solid yellow line; $C_cC_d$ solid red line in Figure 3b) is 1.4 Å; which is a typical aromatic CC bond length [24,80,90,92]. This is in line with the Dewar–Evans–Zimmermann rules that compare the transition state of a symmetry-allowed pericyclic reaction with aromatic cyclopolyenes benefiting from electron delocalization. Next, spin-recoupling occurs, resulting in the formation of a 2-butene structure with some 1,4-biradical character due to pyramidalization of the terminal $CH_2$ groups, with some similarity to distorted ethene (CC bond lengthened and $CH_2$ groups pyramidalized) and the formation of two new CC bonds and the cyclohexene. We have found similar curvature patterns for other symmetry-allowed pericyclic reactions being characterized by small curvature peaks in the entrance channel, due to smaller collective energy saving changes leading to moderate reaction barriers in stark contrast to symmetry-forbidden reactions with large curvature peaks in the entrance channel and higher barriers, such as for the symmetry-forbidden [2 + 2] cycloaddition of HF addition to ethylene [92,174]. There is an almost linear decrease of the $C_aC_f$ in the $C_bC_c$ distance, respectively (see Figure 3b), until in phase 8, the final CC bond distance is reached. As revealed by Figure 3d there is only a small charge transfer from ethylene to butadiene in the entrance channel peaking shortly after M5 at the onset of $C_c$ and $C_f$ rehybriziation, followed by a reverse charge transfer from butadiene to ethylene after the TS. For a more in-depth discussion of the parent DA reaction, the reader is referred to Ref. [80].

Figure 4a shows the corresponding URVA analysis for reaction **R1cw**. As in the case of the parent DA reaction, both bonds are formed after the TS. However, in contrast to the parent DA, the new CC bonds are no longer formed simultaneously, because of the ester substituent on one side of the diene. As revealed by Figure 4a, the new $C_bC_c$ bond on the $CH_2$ end is formed first in phase 5 (supportive component, blue color) followed by the formation of the $C_aC_f$ bond on the ester side in phase 6 (supportive component, red color). The preparation for CC bond formation starts before the TS in phase 3 (energy-consuming event), both $C_bC_c$ and $C_aC_f$ components being resistant and with a dominance of the former component. It is interesting to note that there is a large resisting component of $C_aC_f$ in phase 5, where the $C_bC_c$ bond is formed. Figure 4a shows also the decomposition of the curvature into the pyramidalization angles at carbon atoms $C_a$, $C_b$, $C_c$, and $C_f$ involved in the formation of the new $C_bC_c$ and $C_aC_f$ bonds. According to Figure 4a, pyramidalization of $C_b$ and $C_c$ starts in phase 3 together with the preparation for $C_bC_c$ bond formation. Pyramidalization of $C_a$ and $C_f$ is delayed to phase 5, where the preparation process of the $C_aC_f$ bond formation starts. In summary, Figure 4a shows that the formation of the six-member ring is a step–wise process, where preparation of the bond formation goes hand-in-hand with pyramidalization of the carbon atoms which are supposed to form the new CC single bonds.

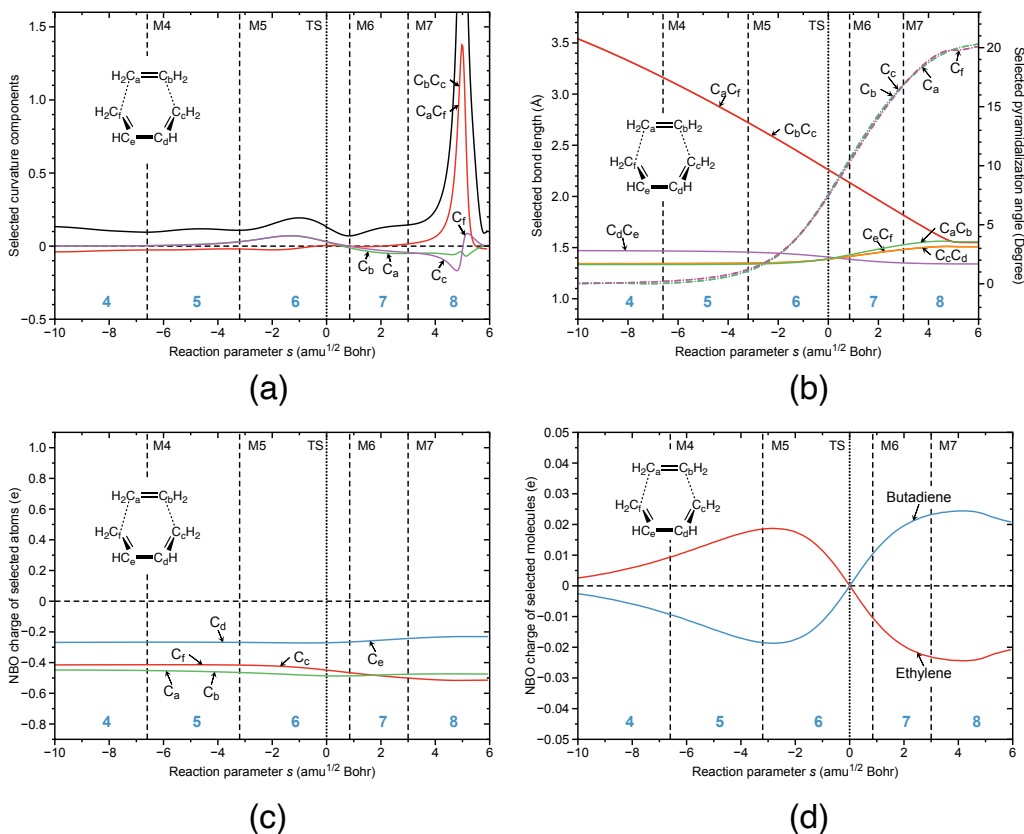

**Figure 3.** Reaction properties along the reaction path for the parent DA reaction; (**a**) decomposition of the reaction curvature into selected components: the bonds shown as two character labels, and the pyramidalization angles shown as a one character label; (**b**) change of the selected geometrical parameters: the bonds shown as two character labels, and the pyramidalization angles shown as a one character label; (**c**) change of the NBO charges of the selected atoms; (**d**) change of the NBO charges of the selected molecules. B3LYP/6-31G(d,p) level of theory.

Figure 4b shows changes in bond lengths along the reaction path for the bonds involved in the six-member ring of the final product. According to Figure 4b, the $C_bC_c$ bond in the reactant complex is already ca 0.4 Å shorter than the corresponding $C_aC_f$ bond, explaining why this bond is formed first. Figure 4b further reveals that during the cycloaddition, the lengths of both bonds continuously decrease. The $C_bC_c$ bond length reaches its final value at the curvature peak of phase 5 indicating on the finalization of $C_bC_c$ bond formation. The $C_aC_f$ bond reaches its final bond length at the curvature peak of phase 6 indicating on the finalization of $C_aC_f$ bond formation. Figure 4b also shows changes of the pyramidalization angles for the carbon atoms $C_a$, $C_b$, $C_c$, and $C_f$ involved in the formation of the new CC bonds. According to Figure 4b, the pyramidalization angles of those four carbon atoms are close to 0 degrees in the reactant complex, and the values of these angles are then increasing, showing how sp$^2$ hybridization of those four carbon atoms changes into the sp$^3$ hybridization from the entrance to the exit channel of the reaction. The values of both of the pyramidalization angles, $C_b$ and $C_c$, start to increase stronger in phase 3 of the reaction, where the preparation for the $C_bC_c$ bond formation starts, and where the $C_bC_c$ bond length starts to decrease more strongly. The stronger increase of the pyramidalization angles $C_b$ and $C_c$ stops in phase 5 of the reaction, where the $C_bC_c$ bond formation is finalized, and where the $C_bC_c$ bond length starts to reach its final value in the product. Similarly, the pyramidalization angles $C_a$ and $C_f$ start to increase strongly in phase 5, where the preparation process for the $C_aC_f$ bond formation takes place, and where the $C_aC_f$ bond length starts to decrease strongly. The strong increase of the pyramidalization angles $C_a$ and

$C_f$ stops in phase 6, where the $C_aC_f$ bond formation takes place, and where the $C_aC_f$ bond length starts to reach its final value in the product. The bond length changes of the other CC bonds involved in ring formation reflect typical changes between single and double bonds; e.g., the bond length of the $C_dC_e$ bond (yellow line) decreases from the entrance to the exit reaction channel, indicating the change from single to double bond in the final product.

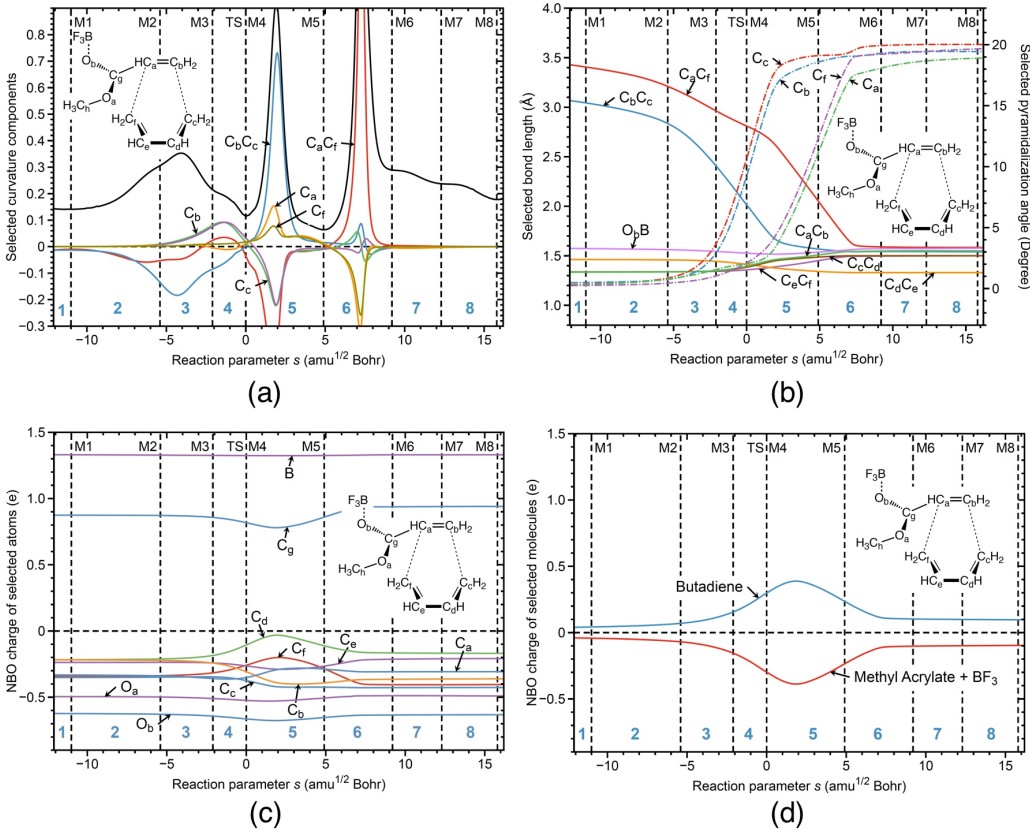

**Figure 4. R1cw** reaction properties along the reaction path; (**a**) decomposition of the reaction curvature into selected components—the bonds are shown as two character labels, and the pyramidalization angles shown as a one character label; (**b**) change of the selected geometrical parameters—the bonds are shown as two character labels, and the pyramidalization angles shown as one character label; (**c**) change of the NBO charges of the selected atoms; (**d**) change of the NBO charges of the selected molecules. B3LYP/6-311+G(2d,p)/AMBER level of theory.

NBO charges along the reaction path are presented in Figure 4c for the selected atoms of the reaction complex. According to Figure 4c, significant charge changes of all carbon atoms involved in the ring formation start in phase 3, which is characterized by the curvature peak due to the preparation of $C_bC_c$ bond formation. According to Figure 4b, in this phase, the $C_bC_c$ bond length strongly decreases while the pyramidalization angles at $C_b$ and $C_c$ strongly increase. The largest charge changes of the all atoms involved in ring formation take place in phase 5, the reaction phase being dominated by the formation of the $C_bC_c$ bond and the preparation process for $C_aC_f$ bond formation. Charge transfer along the reaction path between the butadiene dienophile unit and methyl acrylate diene unit with BF$_3$ attached are presented in Figure 4d. According to Figure 4d, charge transfer from the butadiene unit to the methyl acrylate continuously increases starting from a value of 0.04 e in the reactant complex to a maximum value of 0.39 e in phase 5 and then continuously decreases, reaching its final value of 0.1 e in phase 6. It is important to note that the maximum charge transfer occurs at the curvature peak in phase 5 where the $C_bC_c$ bond is formed, while the final value is reached in phase 6 at the point where the $C_aC_f$ is formed. This shows that charge transfer, pyramidalization, and CC bond formation are

synchronized. In summary, as revealed by Figure 4, the catalyzed reaction in water solution **R1cw** between butadiene and methyl acrylate with $BF_3$ starts in phase 3 with an energy-consuming event preparing for CC bond formation which is characterized by electron density transfer from butadiene to methyl acrylate, changes of the pyramidalization angles of the $C_b$ and $C_c$ carbon atoms, and bond length changes to bring the carbon atoms forming the new CC bonds closer together along the reaction path.

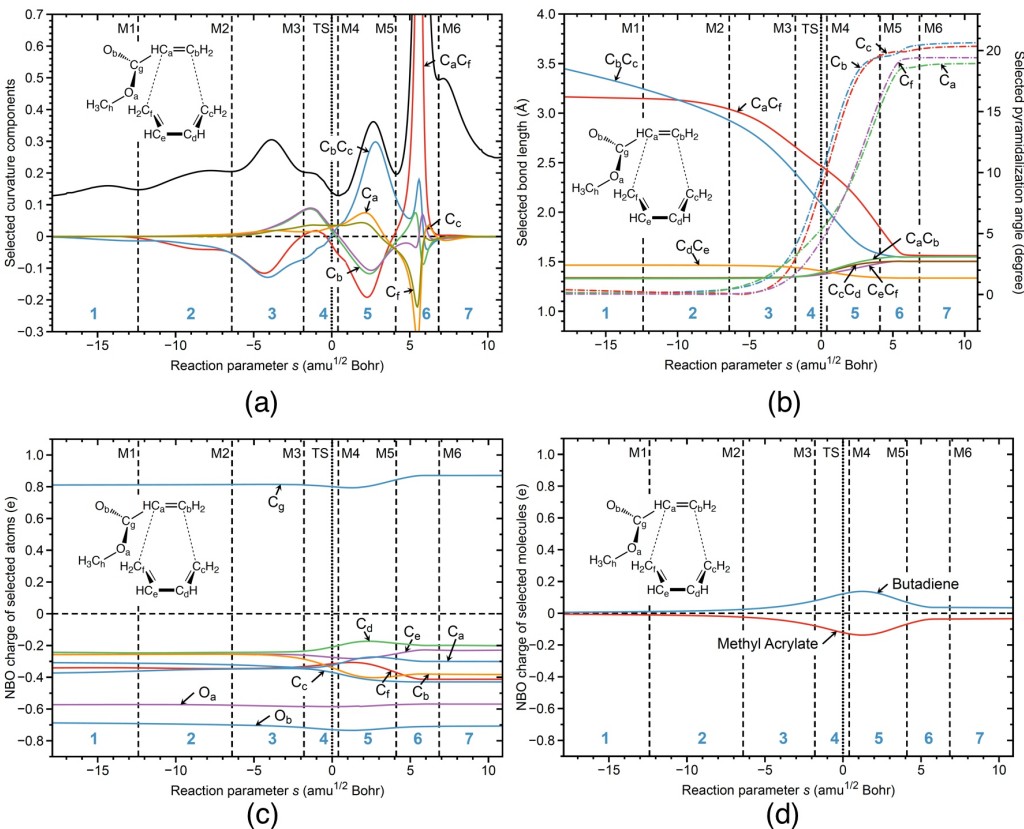

**Figure 5. R1nw** reaction properties along the reaction path; (**a**) decomposition of the reaction curvature into selected components: the bonds shown as two character labels, and the pyramidalization angles shown as a one character label; (**b**) change of the selected geometrical parameters: the bonds shown as two character labels, and the pyramidalization angles shown as a one character label; (**c**) change of the NBO charges of the selected atoms; (**d**) change of the NBO charges of the selected molecules. B3LYP/6-311+G(2d,p)/AMBER level of theory.

In Figure 5, the URVA analysis of the non-catalyzed reaction in water is shown, isolating the catalytic influence of the aqueous environment. Overall, the curvature profile and its decomposition into components for **R1nw** shown in Figure 5a resembles that of the $BF_3$ catalyzed reaction in water **R1cw**. The formation of the $C_bC_c$ and $C_aC_f$ bonds takes place in phases 5 and 6 after TS, with a preparatory phase 3 located before the TS. However, there are two important differences. (i) The peak height of the curvature maximum for $C_bC_c$ bond formation is strongly reduced compared to the catalyzed reaction in water. The height of a curvature peaks is often connected to the necessary effort of a reaction system executing the electronic structure changes associated with bond breaking or bond forming, and reveals the stiffness of the reaction complex with regard to the necessary electronic structure reorganization. Without a catalyst, the ester group can move more freely, facilitating the $C_bC_c$ bond formation. This is also reflected in the bond length changes shown in Figure 5b. In contrast to reaction **R1cw**, reaction **R1nw** starts with a longer $C_bC_c$ distance (3.5 versus 3.15 Å for **R1nw** and **R1cw**, respectively). The changes of the geometrical parameters of the **R1nw** reaction along the reaction path are presented in Figure 5b. In the

staring geometry of the **R1nw** reaction complex, methyl acrylate is shifted away from the butadiene center, making the distance between the $C_a$ and $C_f$ atoms smaller than in the **R1cw** reaction, where the $BF_3$ catalyst orients methyl acrylate closer to the butadiene center by an electrostatic attraction between the negatively charged F atoms of the catalyst and the positively charged H atoms of $C_f$. As the reaction proceeds, the relative lengths of the $C_bC_c$ and $C_aC_f$ bonds are switched in the middle of phase 2, and the preparation process for both bonds starts in phase 3, followed by $C_bC_c$ bond formation in phase 5 and $C_aC_f$ bond formation in phase 6. In comparison to reaction **R1cw**, the overall preparation process of reaction **R1nw** (phases 1–3) is longer and more pronounced, contributing to the energy barrier. The curvature contribution of the pyramidalization angles of atoms $C_a$ and $C_f$ in reaction **R1nw** starts already in phase 3 requiring energy to proceed, in contrast to the reaction **R1cw** where the most significant changes in the pyramidalization of atoms $C_a$ and $C_f$ occur in phase 5 after the TS, and where the reaction path moves already downhill on the PES not contributing to the activation energy.

Figure 5c shows the changes of the NBO charges of the selected atoms for the non-catalyzed reaction **R1nw**, and Figure 5d presents the total charge transfer between the butadiene and methyl acrylate units along the reaction path. The non-catalyzed reaction **R1nw** starts with a zero charge difference between the reacting species, which is different from the catalyzed reaction **R1cw**, where there is already a small charge transfer from butadiene to methyl acrylate caused by the presence of the $BF_3$ catalyst inducing a mutual polarization of butadiene and methyl acrylate. According to Figure 5c, more pronounced charge transfer starts in phase 4 of the reaction **R1nw**, where the average separation between methyl acrylate and butadiene is in a range of 2.5 Å. In contrast, in the catalyzed reaction **R1cw**, significant charge transfer starts already in phase 3, with an average separation between the reacting species in a range of 3.0 Å, indicating on the stronger electron density transfer from butadiene to methyl acrylate, induced by the catalyst. As the non-catalyzed reaction **R1nw** proceeds, the charge difference between butadiene and methyl acrylate increases, reaching the maximum in phase 5 of the reaction. However the charge transfer between the reacting species in the non-catalyzed reaction **R1nw** along the reaction path is overall smaller than that for the catalyzed reaction **R1cw**. In summary, the $BF_3$ catalyst causes already the mutual polarization of butadiene and methyl acrylate in reaction **R1cw** from the start. At the TS of the non-catalyzed reaction **R1nw** in aqueous solution, the charge difference between butadiene and methyl acrylate is 0.27 e compared to a considerably larger value of 0.73 e for the catalyzed reaction **R1cw** in aqueous solution. The stronger charge transfer from butadiene to methyl acrylate in the catalyzed reaction **R1cw** facilitates pyramidalization of the carbon atoms involved in the CC bond formation, a process which starts before the TS and as such contributes to the activation energy. This is reflected in the $E^a$ values of 9.06 kcal/mol for reaction **R1cw** compared to 15.71 kcal/mol for reaction **R1nw**.

Next, URVA results for the catalyzed reaction in the gas phase **R1cg** are presented in Figure 6, singling out the influence of the $BF_3$ catalyst. One obvious difference between the catalyzed reaction in water **R1cw** and the catalyzed reaction in the gas phase **R1cg** is that in the gas phase, the reaction path is substantially longer with regard to both the entrance and exit channel leading to 14 reaction phases compared to 7 phases for the reaction **R1cw**. The aqueous environment confines the free gas phase movements of the reaction partners. For better comparison with the analysis in water, Figure 6 shows the reaction in the range $s = -15$ and $15$ amu$^{1/2}$Bohr corresponding to phases 6 to 14. Figure 6a shows the decomposition of the reaction curvature into components, starting from phase 6. The overall curvature pattern resembles that of the previously discussed reactions, i.e., preparation phases including contributions of the pyramidalization angles of the carbon atoms involved in the formation of the new CC bonds before the TS and staggered CC bond formation, where the $C_bC_c$ bond is formed first followed by the $C_aC_f$ bond formation after the TS. It is interesting to note that the height of the $C_bC_c$ and $C_aC_f$ curvature peaks of reaction **R1cg** is in between the two extremes given by reactions **R1cw** and **R1nw** (i.e., being equal in **R1cw** and quite distinct in **R1nw**) which is reflecting the influence of the catalyst on $C_bC_c$ bond

formation. In the way, the $C_bC_c$ peak can serve in future studies as a fingerprint of the catalyst. The entrance and exit channels of the reaction in the gas phase are longer than those of the reaction in aqueous solution because the six-membered ring can more freely adjust in the gas phase. There is a large peak in phase 18 of the reaction **R1cg**, characterized by conformational changes, e.g., the reorientation of the ester group.

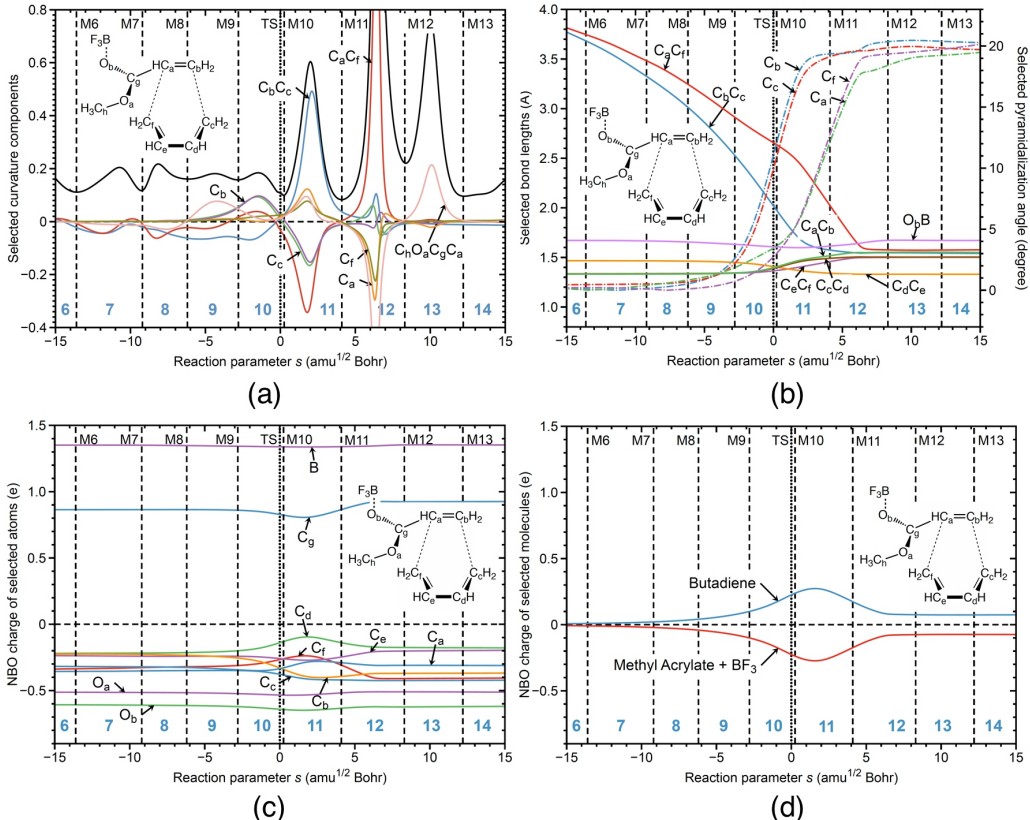

**Figure 6. R1cg** reaction properties along the reaction path; (**a**) decomposition of the reaction curvature into selected components; (**b**) change of the selected geometrical parameters; (**c**) change of the NBO charges of selected atoms; (**d**) change of the NBO charges of selected molecules. B3LYP/6-311+G(2d,p) level of theory.

Figure 6b shows the changes of the geometrical parameters for reaction **R1cg**. The overall pattern is much closer to that of reaction **R1cw** than to reaction **R1nw** with a stronger decrease of the $C_bC_c$ bond distance from the very beginning of the reaction, although the gap between changes of $C_bC_c$ bond and $C_aC_f$ bond distances is smaller. This shows that the aqueous environment supports the catalytic function of $BF_3$. The same holds for the NBO charge changes along the reaction path shown in Figure 6c, and the total charge transfer between the butadiene and methyl acrylate unit shown in Figure 6d. The charge transfer at the TS of reaction **R1cg** is with a value of 0.44 e larger than that of reaction **R1nw** and smaller than that of reaction **R1cw** (0.27 e and 0.73 e, respectively). Overall, the electron density transfer induced by the $BF_3$ catalyst facilitating the pyramidalization of the carbons atoms involved in the new CC bond formation is stronger than that caused by the aqueous environment, which is reflected by the activation energies, 9.06 kcal/mol for reaction **R1cg** versus 15.71 kcal/mol for reaction **R1nw**.

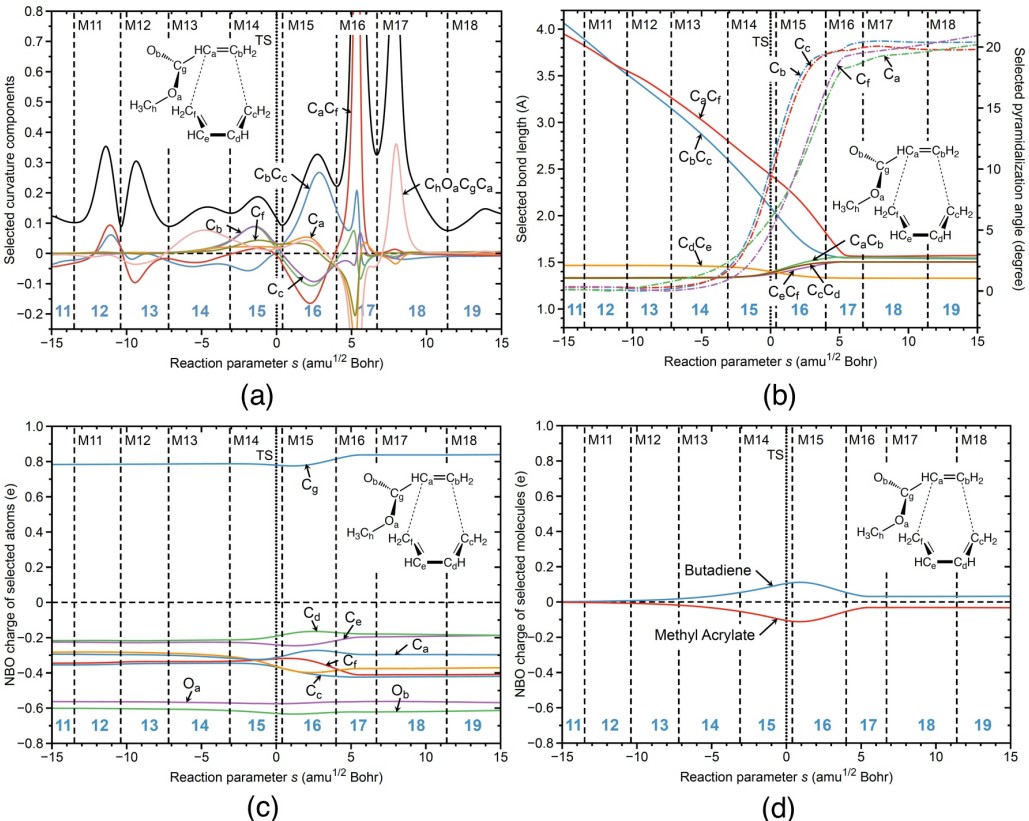

**Figure 7. R1ng** reaction properties along the reaction path; (**a**) decomposition of the reaction curvature into selected components; (**b**) change of the selected geometrical parameters; (**c**) change of the NBO charges of selected atoms; (**d**) change of the NBO charges of selected molecules. B3LYP/6-311+G(2d,p) level of theory.

In Figure 7, the URVA results for the non-catalyzed reaction in the gas phase **R1ng** are shown. As in the case of the gas phase reaction **R1cg**, the reaction path is substantially longer with regard to both the entrance and exit channel, leading to 19 reaction phases compared to 7 phases for reaction **R1cw**. For better comparison with the aqueous solution data, results are presented in the range $s = -15$ and 15 amu$^{1/2}$Bohr corresponding to phases 11 to 19. Figure 7a shows the curvature profile. The curvature pattern connected with the formation of the two CC bonds (phases 16 and 17) strongly resembles that of other noncatalyzed reactions investigated in this work. As already seen for reaction **R1nw** without the catalyst, the $C_bC_c$ bond formation peak is less pronounced than that of the $C_aC_f$ bond formation; also, the contributions of the pyramidalization angles of the carbon atoms involved in the CC bond formation are quite similar in phases 16 and 17 to those of reaction **R1nw**. Comparison with the catalyzed reaction **R1cg** discloses the influence of the aqueous environment. As for reaction **R1cg**, there is an additional large curvature peak in the exit channel in phase 18 dominated by the reorientation of the ester group. The preparation phases 11–15 are more pronounced than the corresponding phases 6–10 in reaction **R1cg**. Important is to note that preparation for CC bond formation starts early in phases 12 and 13, contributing to the energy along the reaction path. As obvious from Figure 7b presenting how the geometrical parameters change along the reaction path, reaction **R1ng** starts with $C_aC_f$ and $C_bC_c$ bond distances in the range of 4 Å compared to 3.7 Å for reaction **R1cg** and 3.2–3.5 Å for reaction **R1cw**. This implies that more energy is requested to decrease the distance between the carbon atoms forming the new CC bonds, a process which is more difficult than in the case of the catalyzed reactions because of the lack of substantial charge transfer. As depicted in Figure 7d, charge transfer between the

butadiene and methyl acrylate units is the smallest found for all reactions, with a value of 0.21 e at the TS compared to 0.27 e for reaction **R1nw** and 0.73 e for **R1cw**. However, the question remains whether these deficiencies alone can explain the activation energy of 20.68 kcal/mol compared to 9.06 kcal/mol for reaction, which is discussed in more detail in the following sections by a comparison of molecular properties evaluated at the stationary points of the four reactions including a bond-strength analysis of the CC bonds being involved in the ring formation, potential hydrogen bonding in the TS, as well as a puckering of the six-membered ring.

### 3.3. Chemical Bond Analysis

Table 2 presents bond distances $d$, local mode force constants $k^a$ and corresponding *BSO* values (describing the bond strength) and local mode frequencies $\omega^a$, as well as the electron density $\rho$ and the energy density $H_\rho$ at the bond critical point (describing the covalent character) for the $C_aC_f$ and $C_bC_c$ bonds evaluated at the stationary points of reactions **R1cw**, **R1nw**, **R1cg**, and **R1ng**. Relations between these properties are graphically displayed in Figure 8.

**Table 2.** The $C_aC_f$ and $C_bC_c$ bond length $d$, the local mode force constant $k^a$, *BSO*, the local mode frequency $\omega^a$, the electron density $\rho$, and the energy density $H_\rho$ at the bond critical point of the stationary points for reaction **R1**. B3LYP/6-311+G(2d,p)/AMBER level of theory in water solution, and B3LYP/6-311+G(2d,p) level of theory in the gas phase.

| Reaction | Bond [1] | Molecule [1] | $d$ | $k^a$ | *BSO* | $\omega^a$ | $\rho$ | $H_\rho$ |
|---|---|---|---|---|---|---|---|---|
| | | | Å | mDyn/Å | | cm$^{-1}$ | e/Bohr$^3$ | Hr/Bohr$^3$ |
| **R1cw** | $C_aC_f$ | Re | 3.431 | 0.048 | 0.029 | 116.7 | 0.0061 | 0.0008 |
| | | TS | 2.810 | 0.113 | 0.058 | 179.1 | 0.0187 | 0.0011 |
| | | Pr | 1.584 | 2.715 | 0.733 | 876.4 | 0.2103 | −0.1407 |
| | $C_bC_c$ | Re | 3.069 | 0.071 | 0.040 | 141.3 | 0.0093 | 0.0013 |
| | | TS | 2.014 | 0.061 | 0.035 | 131.6 | 0.0776 | −0.0225 |
| | | Pr | 1.540 | 3.592 | 0.916 | 1008.0 | 0.2331 | −0.1718 |
| **R1nw** | $C_aC_f$ | Re | 3.164 | 0.117 | 0.060 | 182.0 | 0.0078 | 0.0012 |
| | | TS | 2.465 | 0.120 | 0.061 | 184.1 | 0.0339 | −0.0026 |
| | | Pr | 1.560 | 3.165 | 0.828 | 946.2 | 0.2220 | −0.1563 |
| | $C_bC_c$ | Re | 3.449 | 0.030 | 0.020 | 92.6 | - | - |
| | | TS | 2.088 | 0.082 | 0.045 | 152.5 | 0.0677 | −0.0173 |
| | | Pr | 1.550 | 3.450 | 0.887 | 987.8 | 0.2287 | −0.1644 |
| **R1cg** | $C_aC_f$ | Re | 6.114 | - | - | - | - | - |
| | | TS | 2.657 | 0.127 | 0.064 | 189.8 | 0.0242 | 0.0001 |
| | | Pr | 1.548 | 3.323 | 0.861 | 969.6 | 0.2287 | −0.1659 |
| | $C_bC_c$ | Re | 5.565 | - | - | - | - | - |
| | | TS | 2.020 | 0.084 | 0.046 | 154.2 | 0.0780 | −0.0226 |
| | | Pr | 1.532 | 3.894 | 0.977 | 1049.5 | 0.2388 | −0.1793 |
| **R1ng** | $C_aC_f$ | Re | 5.495 | - | - | - | - | - |
| | | TS | 2.438 | 0.119 | 0.060 | 183.1 | 0.0364 | −0.0035 |
| | | Pr | 1.541 | 3.550 | 0.908 | 1002.1 | 0.2342 | −0.1726 |
| | $C_bC_c$ | Re | 9.456 | - | - | - | - | - |
| | | TS | 2.088 | 0.092 | 0.049 | 161.7 | 0.0691 | −0.0179 |
| | | Pr | 1.533 | 3.873 | 0.973 | 1046.6 | 0.2385 | −0.1788 |

[1] For the bond description, see the corresponding reaction plot. Re—reactant, TS—transition state, Pr—product.

The catalyzed reaction in solution **R1cw** starts from a van der Waals complex (reactant Re in Table 2) with a weak chemical interaction between atoms $C_a$ and $C_f$. The $C_aC_f$ distance has a value of 3.431 Å, which leads to the local mode force constant of a value of 0.048 mDyn/Å and a small *BSO* value of 0.029. The energy density $H_\rho$ has a value of 0.0008 Hartree/Bohr$^3$, classifying this interaction as predominately electrostatic. At TS, the $C_a$ and $C_f$ bond length decreased to a value of 2.810 Å, and the local mode force

constant and corresponding *BSO* values increased accordingly (0.113 mDyn/Å and 0.058, respectively), whereas $\rho$ has slightly increased too (0.0011 Hartree/Bohr³) and the interaction is still predominately electrostatic as reflected by an $H_\rho$ value of 0.0008 Hartree/Bohr³. In the final product of this reaction (Pr in Table 2) the $C_aC_f$ bond has reached a distance d of length 1.584 Å which comes closer to the CC single bond distance of ethane, reflected by the local mode force constant of 2.715 mDyn/Å and the corresponding *BSO* value of 0.733. The $\rho$ value has increased to −0.1407 Hartree/Bohr³ and $H_\rho$ with a value of −0.1407 Hartree/Bohr³ identifies this bond as covalent.

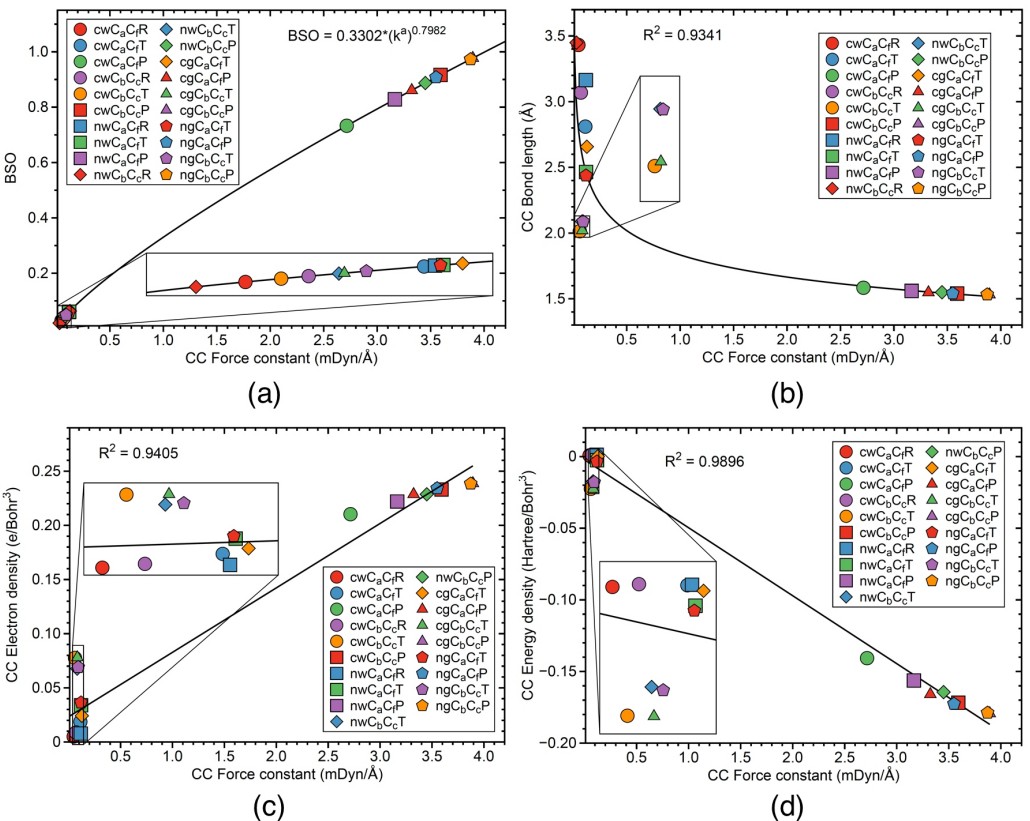

**Figure 8.** The $C_aC_f$ and $C_bC_c$ bond properties at the stationary points of the reaction **R1**; (**a**) *BSO* as a function of the local mode force constant; (**b**) correlation between the bond length and the local mode force constant; (**c**) correlation between the electron density at a bond critical point and the local mode force constant; (**d**) correlation between the energy density at a bond critical point and the local mode force constant. For the bond description, see the corresponding reaction plot. Re—reactant, TS—transition state, Pr—product. B3LYP/6-311+G(2d,p)/AMBER level of theory in water solution, and B3LYP/6-311+G(2d,p) level of theory in the gas phase.

A similar trend is found for the $C_bC_c$ bond in this reaction. In the reactant complex, the length of this bond has a value of 3.069 Å, the force constant has a value of 0.071 mDyn/Å, and the energy density has a value of 0.0013 Hartree/Bohr³, showing a weak electrostatic interaction between the $C_b$ and $C_c$ atoms. At TS, the bond length, the force constant, and the energy density have values of 2.014 Å, 0.061 mDyn/Å, and −0.0225 Hartree/Bohr³, respectively, and in the final product these values are 1.540 Å, 3.592 mDyn/Å, and −0.1718 Hartree/Bohr³, respectively. The $C_bC_c$ bond is slightly stronger than the $C_aC_f$ bond from the beginning as reflected by the bond distances, local mode and electron density properties. The $C_bC_c$ bond distance of 2.810 Å at the TS compared a value of 2.014 Å for the $C_aC_f$ bond shows that $C_bC_c$ formation is ahead of $C_aC_f$ bond formation in line with the curvature profile. This is also reflected in the $H_\rho$ values, whereas the $C_aC_f$ interaction is of electrostatic nature at the TS, it is already covalent for

the $C_bC_c$ bond. The non-catalyzed reaction in solution **R1nw**, starts also from the van der Waals complex, showing similar changes of the $C_aC_f$ and $C_bC_c$ bond properties, as in the catalyzed reaction **R1cw**. Going from the reactant complex, through TS, to the final product, the length of the $C_aC_f$ bond decreases (3.164, 2.465, and 1.560 Å, respectively), the force constant increases (0.117, 0.120, and 3.165 mDyn/Å, respectively), and the energy density becomes more negative (0.0012, −0.0026, and −0.1563 Hartree/Bohr$^3$, respectively). Similarly, the length of the $C_bC_c$ bond decreases (3.449, 2.088, and 1.550 Å, respectively), and the force constant increases (0.030, 0.082, and 3.450 mDyn/Å, respectively). However, there is no critical point for this bond in the reactant complex, and for TS and the final product, the energy density becomes more negative as well (−0.0173 and −0.1644 Hartree/Bohr$^3$, respectively). For the non-catalyzed reaction in solution **R1nw**, we observe similar bond property changes at TS as for the catalyzed reaction **R1cw**. The strength of the $C_aC_f$ bond is bigger than the strength of the $C_bC_c$ bond at TS (0.120 and 0.082 mDyn/Å, respectively), while the energy density of the $C_aC_f$ bond is less negative than in the $C_bC_c$ bond at TS (−0.0026 and −0.0173 Hartree/Bohr$^3$, respectively), which indicates on a more covalent character of the $C_bC_c$ bond, leading also to its smaller bond length (2.465 and 2.088 Å, for the $C_aC_f$ and $C_bC_c$ bond, respectively). The catalyzed reaction in the gas phase **R1cg**, starts at the van der Waals complex, in which the $C_a$ and $C_f$ atoms are separated by a distance of a value of 6.114 Å, while the $C_b$ and $C_c$ atoms are separated by a distance of a value of 5.565 Å. At those interatomic distances, we do not observe in our study a local mode and a bond critical point as well. As the reaction proceeds from TS to the product, the $C_aC_f$ bond length is decreasing (2.657 and 1.548 Å, respectively), the force constant is increasing (0.127 and 3.323 mDyn/Å, respectively), and the energy density becomes more negative (0.0001 and −0.1659 Hartree/Bohr$^3$, respectively). The similar changes are observed for the $C_bC_c$ bond in this reaction: the bond length decreases (2.020 and 1.532 Å, respectively), the force constant increases (0.084 and 3.894 mDyn/Å, respectively), and the energy density becomes more negative (−0.0226 and −0.1793 Hartree/Bohr$^3$, respectively). The strength of the $C_aC_f$ bond is bigger than the strength of the $C_bC_c$ bond at TS (0.127 and 0.084 mDyn/Å, respectively), however the energy density of the $C_aC_f$ bond is less negative than in the $C_bC_c$ bond at TS (0.0001 and −0.0226 Hartree/Bohr$^3$, respectively), which shows that the strength of the $C_bC_c$ bond is more related to the covalent character of this bond, leading to its smaller bond length (2.657 and 2.020 Å, for the $C_aC_f$ and $C_bC_c$ bonds, respectively). Similar to the catalyzed reaction in the gas phase **R1cg**, the non-catalyzed gas phase reaction **R1ng** starts from the reactant complex with a large interatomic distance between the $C_a$ and $C_f$ atoms (5.495 Å), as well as between the $C_b$ and $C_c$ atoms (9.456 Å), with no local modes and bond critical points being observed. The changes of the $C_aC_f$ and $C_bC_c$ bond parameters, going from TS to the product in the **R1ng** reaction, are similar to the change in the **R1cg** reaction. The $C_aC_f$ bond length decreases (2.438 and 1.541 Å, respectively), and the $C_bC_c$ bond length decreases (2.088 and 1.533 Å, respectively). The $C_aC_f$ force constant is increasing (0.119 and 3.550 mDyn/Å, respectively), and the $C_bC_c$ force constant is increasing (0.092 and 3.873 mDyn/Å, respectively). The $C_aC_f$ energy density becomes more negative (−0.0035 and −0.1726 Hartree/Bohr$^3$, respectively), and the $C_bC_c$ energy density becomes more negative (−0.0179 and −0.1788 Hartree/Bohr$^3$, respectively). At TS, the strength of the $C_aC_f$ bond is larger than the strength of the $C_bC_c$ bond (0.119 and 0.092 mDyn/Å, respectively), however the $C_aC_f$ energy density is less negative than the $C_bC_c$ energy density (−0.0035 and −0.0179 Hartree/Bohr$^3$, respectively), which indicates that although the $C_bC_c$ bond is weaker than the $C_aC_f$ at TS, its strength is related to the more covalent character of this bond.

Figure 8 shows the relations between the properties of the $C_aC_f$ and $C_bC_c$ bonds, for the stationary points of the **R1cw**, **R1nw**, **R1cg**, and **R1ng** reactions. Figure 8a, presents the power relationship between *BSO* and the local mode force constants of the $C_aC_f$ and $C_bC_c$ bonds. The power relationship between these two bond quantities was obtained from the two reference molecules, for which details are provided in the computational details

section of this study, and the *BSO* value of 1.0 corresponds to the strength of the single CC bond in ethane. According to Figure 8a, all $C_aC_f$ and $C_bC_c$ bonds investigated in our study have the strength smaller than the strength of the single CC aliphatic bond. There are two groups of the $C_aC_f$ and $C_bC_c$ bonds, namely the group of the strong bonds corresponding to the products of the reactions having the local mode force constant in a range between 2.7 to 3.9 mDyn/Å, and the group of the weak bonds corresponding to TS and the reactants having the local mode force constant smaller than 0.2 mDyn/Å. Among the bonds in the products, the stronger bonds are observed in our calculations for the gas phase reactions rather than in the solution reactions, and the $C_bC_c$ bond is generally stronger than the $C_aC_f$ bond. According to Table 2, the strength of the $C_aC_f$ bond in the products of the gas phase reactions (3.323 and 3.550 mDyn/Å, for the **R1cg** and **R1ng** reactions, respectively), is bigger than the strength of this bond in the solution reactions (2.715 and 3.165 mDyn/Å, for the **R1cw** and **R1nw** reactions, respectively). Similarly, the strength of the $C_bC_c$ bond in the products of the gas phase reactions (3.894 and 3.873 mDyn/Å, for the **R1cg** and **R1ng** reactions, respectively), is bigger than the strength of this bond in the solution reactions (3.592 and 3.450 mDyn/Å, for the **R1cw** and **R1nw** reactions, respectively).

Overall, the stronger bonds of the products in the gas phase reactions, are consistent with the larger reaction barriers of these reactions. According to Table 1, the reactions in the gas phase are more exothermic (reacion energies of $-34.26$ and $-34.61$ kcal/mol, for the **R1cg** and **R1ng**, respectively), than the reactions in solution ($-30.56$ and $-33.24$ kcal/mol, for the **R1cw** and **R1nw**, respectively). Figure 8b presents the correlation between the $C_aC_f$ and $C_bC_c$ local mode force constants and the bond lengths. Although there is some scattering of the points for the small value of the local mode force constant, the correlation between these two bond properties is reaching the $R^2$ value of 0.9341, showing a general trend that a stronger bond has a smaller bond length in our study. Figure 8c shows the correlation between the local mode force constants and the electron density at the bond critical points (the $R^2$ value of 0.9405), while Figure 8d presents the correlation between the local mode force constants and the energy density at the bond critical points (the $R^2$ value of 0.9896). According to Figure 8d, the stronger $C_aC_f$ and $C_bC_c$ bonds are characterized as bonds with a more covalent character.

*3.4. Hydrogen Bonds*

In order to further investigate the catalytic effect of the aqueous environment, i.e., to determine if the transition states of the reaction in water can benefit from hydrogen bonding decreasing the activation energy, we analyzed reactant Re and TS of the non-catalyzed reaction in water solution **R1nw**. According to our calculations, there are three hydrogen bonds formed between the oxygen atoms of methyl acrylate and hydrogen atoms of solvent water molecules as shown in Figure 9. One hydrogen bond (**HB1**) is formed between the oxygen atom of the acrylate ester group and a water molecule, and two hydrogen bonds are formed between the oxygen atom of the acrylate carbonyl group and two water molecules (**HB2** and **HB3**), and the local mode parameters of those three hydrogen bonds are presented in Table 3. According to Table 3, all hydrogen bonds in TS are shorter and stronger than the corresponding hydrogen bonds in the reactant complex. **HB2** is the strongest hydrogen bond observed in our study, and in TS it has a length of a value of 1.8800 Å and a local mode force constant of a value of 0.277 mDyn/Å. The same hydrogen bond in the reactant complex has a length of a value of 1.9251 Å, and the force constant of a value of 0.202 mDyn/Å, showing a substantial increase of the hydrogen bond strength in TS. The strength of the hydrogen bond **HB1** is increased more than twice going from the reactant to TS (the $k^a$ value of 0.079 and 0.196 mDyn/Å, for the reactant and TS, respectively) and similar increase is observed for the **HB3** hydrogen bond (the $k^a$ value of 0.040 and 0.124 mDyn/Å, for the reactant and TS, respectively). The stronger hydrogen bonds at TS relative to the reactant complex in **R1nw**, is due to a more negative charge of the oxygen atoms in methyl acrylate. The NBO atomic charge of the $O_a$ atom at TS has a value of $-0.585$ e, while the charge of the $O_b$ atom has a value of $-0.730$ e. In the reactant

complex, the atomic charges of the $O_a$ and $O_b$ atoms have values of $-0.572$ and $-0.688$ e, respectively. Therefore, we can conclude that the stronger hydrogen bonds formed with water by the reaction complex at TS, relative to the reactant, contribute to the lowering of the activation energy in water solution, because for the non-catalyzed reaction in the gas phase **R1ng**, there are no hydrogen bonds at TS and the reactant complex, which could stabilize these stationary points differently; therefore, the activation energy of the gas phase reaction **R1ng** is higher than in the reaction in solution **R1nw**.

**Figure 9.** Local mode parameters of hydrogen bonds for the reactant and TS of the reaction **R1nw**. The bond lengths (Å) are shown in a normal font, the local mode force constants (mDyn/Å) are shown in an italic font. B3LYP/6-311+G(2d,p)/AMBER level of theory.

**Table 3.** Hydrogen bond length $d$, local mode force constant $k^a$, *BSO*, and NBO atomic charge on oxygen for the reactant and TS of the reaction **R1nw**. B3LYP/6-311+G(2d,p)/AMBER level of theory.

| H–Bond | Reactant | | | | TS | | | |
| | $d$ | $k^a$ | *BSO* | *O Charge* | $d$ | $k^a$ | *BSO* | *O Charge* |
| | Å | mDyn/Å | | e | Å | mDyn/Å | | e |
|---|---|---|---|---|---|---|---|---|
| HB1 | 2.0363 | 0.079 | 0.237 | $-0.572$ | 1.9283 | 0.196 | 0.312 | $-0.585$ |
| HB2 | 1.9251 | 0.202 | 0.315 | $-0.688$ | 1.8800 | 0.277 | 0.346 | $-0.730$ |
| HB3 | 2.0988 | 0.040 | 0.193 | $-0.688$ | 1.9415 | 0.124 | 0.272 | $-0.730$ |

*3.5. Puckering Analysis*

This section is devoted to a ring-puckering analysis. The contributions of the chair, boat, and tboat form of the six-member ring involved in the reaction complexes of the **R1cw**, **R1nw**, **R1cg**, and **R1ng** reactions, are presented in Table 4. According to Table 4, for the reactant of catalyzed reaction in solution **R1cw**, the biggest contribution is from the boat form (74.7%), with a small addition of the tboat form (24.1%). At TS, the boat contribution is increased (88.2%), while in the final product the boat contribution drops down (69.4%), while the tboat contribution increases (29.3%). For the non-catalyzed reaction in solution **R1nw**, the boat contribution dominates (93.0, 99.3, and 96.1% for the reactant, TS, and the product, respectively). For both the catalyzed **R1cg** and the non-catalyzed **R1ng** reaction in the gas phase, the reactant complexes do not form structures involving the six-member ring. At the TS of the **R1cg** reaction, the contribution of the boat form has a value of 99.1%, while the product of this reaction has a structure involving 39.8% of the chair form, and 60.1% of the tboat form. The similar deformations of the six-member ring are observed in our calculations of the **R1ng** reaction. At TS, the contribution of the boat form has a value of 100%, while in the product, the contribution of the chair and tboat forms have values of 39.8% and 59.4%, respectively.

According to Table 4, for both reactions in solution **R1cw** and **R1nw**, the dominant contribution to the six-member ring puckering is from the boat form, which indicates that the water environment leads to space confinement of the reaction complex, conserving the

boat form of the six-member ring from the entrance to the exit reaction channel. The space confinement is stronger for the non-catalyzed reaction **R1nw**, where the contributions of the boat form to the structure of the six-member ring are always larger than 90%. For the both reactions in the gas phase **R1cg** and **R1ng**, the boat form of the six-member ring dominates only at TS, while the ring is deformed in the final products to a tboat form with some contribution of the chair form. These results are inline with our observation that the gas phase reactions are characterized by longer reaction paths with more phases in both the entrance and exit channel.

**Table 4.** Contribution (%) of the chair, boat, and tboat forms of the six-membered ring at the stationary points of the reaction **R1**. B3LYP/6-311+G(2d,p)/AMBER level of theory in water solution, and B3LYP/6-311+G(2d,p) level of theory in the gas phase.

| Reaction | Molecule [1] | Chair | Boat | Tboat |
|---|---|---|---|---|
| **R1cw** | Re | 1.2 | 74.7 | 24.1 |
|  | TS | 0.6 | 88.2 | 11.2 |
|  | Pr | 1.4 | 69.4 | 29.2 |
| **R1nw** | Re | 1.9 | 93.0 | 5.1 |
|  | TS | 0.1 | 99.3 | 0.6 |
|  | Pr | 0.4 | 96.1 | 3.5 |
| **R1cg** | Re | - | - | - |
|  | TS | 0.0 | 99.1 | 0.8 |
|  | Pr | 39.8 | 0.1 | 60.1 |
| **R1ng** | Re | - | - | - |
|  | TS | 0.0 | 100.0 | 0.0 |
|  | Pr | 39.8 | 0.7 | 59.4 |

[1] Re—reactant, TS—transition state, Pr—product.

## 4. Conclusions

In this study we investigated the DA reaction of methyl acrylate and butadiene, both in the gas phase and aqueous solution, both non-catalyzed and catalyzed by the $BF_3$ Lewis acid using URVA and LMA as major analysis tools, complemented with NBO, electron density and ring-puckering analyses. We considered four different starting orientations of methyl acrylate and butadiene, which led to 16 DA reactions in total. The reaction path curvature profiles exhibit an overall similar pattern for all 16 reaction with the same sequence of CC single bond formation for every reaction. In contrast to the parent DA reaction with symmetric substrates causing a synchronous bond formation process, here, the new CC single bond on the $CH_2$ side of methyl acrylate is formed first, followed by the CC bond at the ester side. As for the parent DA reaction, both bond formation events occur after the TS, i.e., they do not contribute to the energy barrier. What determines the barrier is the preparation process for CC bond formation, including the approach diene and dienophile, CC bond length changes and in particular rehybridization of the carbon atoms involved in the formation of the cyclohexene ring. This process is modified by both the $BF_3$ catalyst and the water environment, where both work in a hand-in-hand fashion leading to the lowest energy barrier of 9.06 kcal/mol found for catalyzed reaction **R1** in aqueous solution compared to the highest energy barrier of 20.68 kcal/mol found for the non-catalyzed reaction **R1** in the gas phase. The major effect of the $BF_3$ catalyst is the increased mutual polarization and the increased charge transfer between methyl acrylate and butadiene, facilitating the approach of diene and dienophile and the pyramidalization of the CC atoms involved in the ring formation, which leads to a lowering of the activation energy. The catalytic effect of water solution is is threefold. The polar environment leads also to increased polarization and charge transfer between the reacting species, similar as in the case of the $BF_3$ catalyst, although to a smaller extend. More important is the formation of hydrogen bonds with the reaction complex, which are stronger for the TS than for the reactant, thus stabilizing the TS which leads to a further reduction of the activation energy. As shown by the ring puckering analysis, the third effect of water is space confinement of

the reacting partners conserving the boat form of the six–member ring from the entrance to the exit reaction channel. In summary, URVA combined with LMA has led to a clearer picture on how both $BF_3$ catalyst and aqueous environment are synchronized, providing valuable new insights into the DA reaction of methyl acrylate and butadiene and beyond.

**Supplementary Materials:** The following are available online at https://www.mdpi.com/article/10.3390/catal12040415/s1: Description of the ring puckering analysis, structures of reactants, TS, and products for all 16 reactions investigated in this work, Figures S1–S16; description of reaction video files for all reactions investigated in this work, Table S1; HOMO-LUMO energy gap at TS for all reactions investigated in this work, Table S2; URVA analysis plots for reactions **R2cw**, **R3cw**, **R4cw**, **R2nw**, **R3nw**, **R4nw**, **R2cg**, **R3cg**, **R4cg**, **R2ng**, **R3cg**, and **R4cg** not shown in the manuscript, Figures S17–S28; reaction movie files for the 16 reactions in mpg format.

**Author Contributions:** Conceptualization, E.K. and M.F.; methodology, M.F. and E.K.; validation, E.K. and M.F.; formal analysis, M.F. and E.K.; investigation, M.F. and E.K.; resources, E.K.; data curation, M.F.; writing—original draft preparation, M.F.; writing—review and editing, E.K.; funding acquisition, E.K. All authors have read and agreed to the published version of the manuscript.

**Funding:** This research was funded by the National Science Foundation NSF, grant CHE 2102461.

**Acknowledgments:** This work was supported by supported by the National Science Foundation, Grant CHE 2102461. We thank the Center for Research Computation at SMU for providing generous high-performance computational resources.

**Data Availability Statement:** All data supporting the results of this work are presented in tables and figure of the manuscript and in the Supplementary Materials.

**Conflicts of Interest:** The authors declare no conflict of interest.

### Abbreviations

The following abbreviations are used in this manuscript:

| | |
|---|---|
| URVA | Unified Reaction Valley Approach |
| LMA | Local Mode Analysis |
| QM/MM | Quantum Mechanical Molecular Mechanical |
| DFT | Density Functional Theory |
| FMO | Frontier Molecular Orbitals |
| HOMO | Highest Occupied Molecular Orbital |
| LUMO | Lowest Unoccupied Molecular Orbital |
| IRC | Intrinsic Reaction Coordinate |
| CNM | Characterization of Normal Mode |
| NBO | Natural Bond Orbital |
| BSO | Bond Strength Order |
| TS | Transition State |

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
