# Peer review of "BF3–Catalyzed Diels–Alder Reaction between Butadiene and Methyl Acrylate in Aqueous Solution—An URVA and Local Vibrational Mode Study"

_catalysts, doi:10.3390/catal12040415_

Round 1
Reviewer 1 Report
In the present contribution, the authors studied the Diels–Alder reaction between methyl acrylate and butadiene catalyzed by BF3. To this end, they applied a number of topological methods as well as URVA and local mode analysis. In general, the study is well-performed and provides a detailed mechanistic insight into the nature of catalysis in these systems.
I have only one serious concern: actually, the activation barriers and thermodynamics are the most important numbers that determine the rate constants. The B3LYP energies are well-known to be skewed for many systems. It is thus reasonable to refine the single-point energies with, e.g., DLPNO-CCSD(T) techniques. They are quite doable in the present case.
Author Response
Reviewer 1
I have only one serious concern: actually, the activation barriers and thermodynamics are the most important numbers that determine the rate constants. The B3LYP energies are well-known to be skewed for many systems. It is thus reasonable to refine the single-point energies with, e.g., DLPNO-CCSD(T) techniques. They are quite doable in the present case.
Following the suggestion of this reviewer the energetics of the investigated reactions in the gas phase were recalculated by performing single point energy calculations with the DLPNO-CCSD(T)/def2-TZVP level of theory based on the optimal geometries from the DFT model chemistry and applying thermochemical corrections from the DFT model chemistry. Table 1 has been updated to include these results.
Reviewer 2 Report
The authors investigated reaction mechanisms of the BF3-catalyzed Diels-Alder (DA) reaction of butadiene with methyl-acrylate in aqueous solution using Unified Reaction Valley Approach, local vibrational mode analysis, and other sophisticated analytical tools. They examined four different orientations of BF3 catalysis and four different reaction systems for the existence of BF3 and surrounding water molecules or not and verified how catalysis and water environment promote the DA reaction. They concluded that the BF3 catalysis and aqueous environments promoted charge transfer between the reacting species, facilitating the approach of diene and dienophile and the pyramidalization of the relevant carbon atoms, which reduces the activation energy. The aqueous solutions further contributed to reducing the activation energy by the stabilization of TS and confining the six-membered ring to the boat form during the reaction process. The manuscript is well organized and describes the research procedures and the results in detail. The topic is of interest for the readership of Catalysts; therefore, I suggest publication after considering the following points.
1) In the URVA analysis, the reaction path curvature was calculated along the IRC, which was then decomposed into internal coordinate components, and the IRC was divided into segments of chemical events based on the results. Dynamic effects such as vibrational excitation are also important with respect to reaction path curvature. I suggest that some consideration be given to the dynamics effects caused by such reaction pathway curvature in the present case.
2) The authors compared the activation energies for four different orientations of methyl acrylate and butadiene and chose R1 to give the lowest activation energy, but there was no discussion as to why R1 has the lowest barrier. Is it possible to explain this by the difference in energy and orbital shape of the HOMO and LUMO?
3) Figure 3 should be explained more carefully to better understand the subsequent URVA results. I guess that the several lines in Fig. 3a are overlapped due to the symmetry of the parent DA reaction system. Without explanations, the reader who is not familiar with theoretical analysis may be confused. In line 207, the authors state that “at curvature minimum M6 the six CC bond lengths are equal (R(CC)=1.4 Å”). Which graphs give us such kind of information? I presume this is from Figure 3b, but it looks like only four line plots around 1.4 Å at M6 (s~1 amu1/2 Bohr) in Fig. 3b. The authors should clarify which figures correspond to the discussions in the manuscript, in the first paragraph of section 3.2.
4) There are several typos in the manuscript. I suggest that you should recheck the entire manuscript again to make it reader-friendly.
Author Response
Reviewer 2
In the URVA analysis, the reaction path curvature was calculated along the IRC, which was then decomposed into internal coordinate components, and the IRC was divided into segments of chemical events based on the results. Dynamic effects such as vibrational excitation are also important with respect to reaction path curvature. I suggest that some consideration be given to the dynamics effects caused by such reaction pathway curvature in the present case.
The calculations in aqueous solution presented in this study are based on MD simulations which were performed at 300K, followed by dynamics simulated annealing to 0K, and followed by QM/MM calculations performed at 0K. Therefore, the results of these calculations do not include the dynamics effects, similarly as the calculations in the gas phase. We are currently working on a computational protocol, which will include a series of QM/MM calculations based on randomly selected snapshots from MD simulations (without simulated annealing), and the results of the calculations will be reported statistically. However, this computational protocol is time consuming because it requires a large set of selected snapshots covering different local minima on the potential energy surface from MD. These results will be presented in a future publication.
The authors compared the activation energies for four different orientations of methyl acrylate and butadiene and chose R1 to give the lowest activation energy, but there was no discussion as to why R1 has the lowest barrier. Is it possible to explain this by the difference in energy and orbital shape of the HOMO and LUMO? We have calculated the HOMO-LUMO energy gaps at TSs for all reactions using the B3LYP/6-311+G(2d,p) level of theory in the gas phase, and the B3LYP/6-311+G(2d,p)/AMBER level of theory in solution. The results are presented in Table S2 of the Supporting Material. According to Table S2, the energy gap for the catalyzed reactions in both gas phase and in aqueous solution, is generally larger than the energy gap for the corresponding non-catalyzed reactions, which is also consistent with the difference in the activation energies of the catalyzed and non-catalyzed reactions with one exception, the catalyzed reaction in solution (R1cw). As explained in the manuscript the major effect of the BF3 catalyst is an increased mutual polarization and increased charge transfer between methyl acrylate and butadiene, facilitating the approach of diene and dienophile and the pyramidalization of the CC atoms involved in the ring formation, which leads to a lowering of the activation energy, supported by a polar environment.
Figure 3 should be explained more carefully to better understand the subsequent URVA results. I guess that the several lines in Fig. 3a are overlapped due to the symmetry of the parent DA reaction system. Without explanations, the reader who is not familiar with theoretical analysis may be confused. In line 207, the authors state that “at curvature minimum M6 the six CC bond lengths are equal (R(CC)=1.4 AÌŠ”). Which graphs give us such kind of information? I presume this is from Figure 3b, but it looks like only four line plots around 1.4 AÌŠ at M6 (s~1 amu1/2 Bohr) in Fig. 3b. The authors should clarify which figures correspond to the discussions in the manuscript, in the first paragraph of section 3.2.
The text related to Figure 3 has been updated and extended according to the comments of the reviewer.
There are several typos in the manuscript. I suggest that you should recheck the entire manuscript again to make it reader-friendly.
All typos were corrected.